Contribution of socio-demographic factors in prevalence of soil-transmitted helminth infections among newly arrived laborers in Jeddah, Saudi Arabia

Wakid Majed H. mwakid@kau.edu.sa 1 2
Al-Refai Mohammad F. 1
1 Department of Medical Laboratory Sciences, Faculty of Applied Medical Sciences, King Abdulaziz University , Jeddah , Saudi Arabia
2 Special Infectious Agents Unit, King Fahd Medical Research Center, King Abdulaziz University , Jeddah , Saudi Arabia
Siddiqui Arif
Electronic publication date: 2024 Oct 8
Publication date: 2024
Volume: 12
Electronic Location ID: e18216
Received 2024 Jul 4; Accepted 2024 Sep 11
Copyright: ©2024 Wakid and Al-Refai
Copyright year: 2024
Copyright holder: Wakid and Al-Refai
License: This is an open access article distributed under the terms of the Creative Commons Attribution License, which permits unrestricted use, distribution, reproduction and adaptation in any medium and for any purpose provided that it is properly attributed. For attribution, the original author(s), title, publication source (PeerJ) and either DOI or URL of the article must be cited.
License URL: https://creativecommons.org/licenses/by/4.0/

Keywords: Prevalence, Soil-transmitted helminths, Jeddah, Saudi Arabia, Socio-demographic, Expatriate laborers, Infection

Funding: The Deputyship for Research & Innovation, Ministry of Education in Saudi Arabia through the project number (IFPRC-045-142-2020) and King Abdulaziz University, DSR, Jeddah, Saudi Arabia IFPRC-045-142-2020 The authors received funding from the Deputyship for Research & Innovation, Ministry of Education in Saudi Arabia through the project number (IFPRC-045-142-2020) and King Abdulaziz University, DSR, Jeddah, Saudi Arabia. The funders had no role in study design, data collection and analysis, decision to publish, or preparation of the manuscript.

==============================
Background

Soil-transmitted diseases are caused by intestinal worms transmitted via various routes into the human body resulting in various clinical manifestations. This study aimed to investigate the socio-demographic factors contributing mainly to the prevalence of soil-transmitted helminths (STHs) among newly arrived laborers in Jeddah, Saudi Arabia.

Methods

A total of 188 stool specimens were collected and examined macroscopically and microscopically using different techniques. In addition, real-time PCR was used as a molecular tool to detect several STHs. The descriptive analysis was used to evaluate demographic data and categorical variables in association with STHs infection.

Results

Of all participants, the mean age was 26.08 ± 2.67 years, including 103 females and 85 males. A total of 86 (45.70%) were illiterate, followed by 60 (31.90%) of participants belonging to elementary level. STHs and other intestinal parasites were detected among 35 newly arrived laborers (18.61%). Nineteen cases (10.11%) were infected with STHs, including 15 cases with one STHs (four A. lumbricoides, four N. americanus, four T. trichiura, two S. stercoralis, one A. duodenale) and four cases with two STHs (two T. trichiura and N. americanus; one S. stercoralis and A. duodenale; one T. trichiura and S. stercoralis). High rates of STHs infection were detected among drivers (11.70%) followed by housemaids (11.20%), but with no significant association to occupation. Higher rates of STHs infection were detected among those used to walk on soil barefoot in their home countries before arriving in Jeddah.

Conclusion

This study found that none of the newly arrived expatriate laborers in Jeddah was aware of STHs. About 10% of them had infection with one or two STHs, mainly among newly arrived laborers from Asian countries. There is a need for awareness programs and regular screening for STHs and other intestinal parasites among newly arrived expatriate laborers.

Introduction

Approximately 50% of the population in tropical and subtropical areas suffer from intestinal parasitic infections associated with helminth and protozoa (Ahmed, 2023). The World Health Organization (WHO) considers infections caused by soil-transmitted helminths (STHs) among the common human diseases and estimates that 1.5 billion people worldwide are infected with at least one or more species of STHs, representing 24% of the world population. These parasites are common in areas with poor sanitation and hygiene, especially in Africa, South America and Asia, where ideal climatic conditions for egg survival and larval development in the environment facilitate transmission (World Health Organization, 2023a).

The main STHs that inhabit the intestine of humans are Ascaris lumbricoides, Trichuris trichiura, Necator americanus, Ancylostoma duodenale and Strongyloides stercoralis (World Health Organization, 2020a; Wakid, 2020a). The first four parasites were given more attention and are considered as a group because they have similar diagnostic procedures and respond to the same medications. N. americanus and A. duodenale are referred together as “hookworms”. Strongyloidiasis is among the most neglected tropical diseases but has recently received increased attention in accordance with WHO 2030 global target (World Health Organization, 2020b).

There is no direct infection from freshly passed stool, or from person-to-person, as the eggs need 2–4 weeks to mature and become infectious. Transmission is mainly through ingestion of mature infective egg in contaminated soil, vegetables, or water (A. lumbricoides, T. trichiura), or via direct penetration of skin by filariform larvae (N. americanus, A. duodenale and S. stercoralis), when walking barefoot on contaminated soil. A. duodenale can also be transmitted by ingestion of filariform larvae (World Health Organization, 2023a; Wakid, 2020a; World Health Organization, 2020c). It is estimated that A. lumbricoides, T. trichiura, hookworms and S. stercoralis infect approximately 1 billion, 800 million, 700 million and 70 million, respectively (Holland et al., 2022; Truscott, Turner & Anderson, 2015; World Health Organization, 2013).

Heavy infection with STHs usually cause a range of health symptoms such as abdominal pain, weakness, diarrhea, dysentery, blood and protein loss, anemia, inflammation, rectal prolapse, physical and cognitive growth retardation, intestinal obstruction, impairment of nutrient intake, digestion and absorption. In case of reduced host immunity, S. stercoralis can cause hyperinfection/dissemination syndrome that is often fatal despite the treatment (World Health Organization, 2023a; World Health Organization, 2023b).

According to WHO, morbidity can be reduced intensively in endemic areas through periodic preventive chemotherapy to people at risk. In addition: health and hygiene education reduce transmission and reinfection by encouraging healthy behaviors and improvement in sanitation, but not always possible in resource-poor settings (World Health Organization, 2023b; World Health Organization, 2021).

Global changes in the human sociocultural spectrum have altered the parasitic relationship of worms with humans (Alum, Rubino & Ijaz, 2010; World Health Organization, 2006). Income or poverty are socio-economic characteristics. Agriculture and fishing (Nobre et al., 2013), the size of the family and number of siblings (Bouwmans et al., 2016), the age of the children (Okpala et al., 2014; Jacobsen et al., 2007), the educational standing of the household (Heidari & Rokni, 2003), and the health monitoring or provision of health education (Fouamno Kamga et al., 2011; Kanoa et al., 2006), all have a role in the parasitic disease’s continuous transmission.

In our previous study in Jeddah, we investigated the prevalence of intestinal parasites, and compared several techniques for the detection of STHs among newly arrived expatriate laborers (Al-Refai & Wakid, 2024). The current study looked at the socio-demographic conditions as a contributing factor in the prevalence of STHs among those laborers.

Materials and Methods

This study was conducted according to the guidelines of the Declaration of Helsinki and approved by The Ethics and Research Committee of the Faculty of Applied Medical Sciences, King Abdulaziz University, (FAMS-EC2018-012), and written informed consent was obtained from each participant.

This cross-sectional study included newly arrived expatriate laborers to Jeddah, during the week prior to specimens’ collection, without any exclusion criteria, and regardless of age, gender and nationality. A total of 188 stool specimens were collected and investigated at parasitology laboratory as previously illustrated in details in the first part of our study (Al-Refai & Wakid, 2024), including macroscopic examination (to detect stool consistency, color, presence of blood, mucus, and any adult worms or gravid segments of tapeworms), direct wet smears (using saline and iodine to examine 1–2 mg of stool), sedimentation formal ether technique (a concentration method to examine 2 g of stool), permanent staining with trichrome (to confirm the morphological features of the intestinal protozoan amoebas and flagellates), modified Kinyoun’s (to detect Cryptosporidium oocyst), rapid immunochromatographic diagnostic tests (specific to detect Giardia lamblia, Cryptosporidium, and Entamoeba histolytica antigens in stool), and real-time PCR using specific primers for STHs (target genes included ITS1 for A. lumbricoides and T. trichiura, ITS2 for A. duodenale and N. americanus, and 18S rRNA for S. stercoralis).

A verbal interview was conducted with each newly arrived participant to fill in a questionnaire form and sign an informed consent form. Some of the laborers were from Arab countries and share authors same language. Most of non-Arab laborers were able to understand English, and for those who do not speak or understand Arabic or English, we asked other workers of the same nationality who speaks English or Arabic to help in translation. Demographic data and other information were collected, including gender, age, marital status, nationality, education level, awareness about STHs, walking on soil barefoot, washing vegetables and fruits before eating, and type of residence in their home countries before arriving to Jeddah.

Data were analyzed by using SPSS (version 25; Armonk, NY, USA). Categorical data were reported as frequency, cross-tabulation and percentage. Statistical significance for the variation in the frequency between groups was determined by the Pearson chi-square χ2 test. Relative risk regression analysis (RR) was performed for each investigated factor. The p-value was calculated at a significance level of p < 0.05.

Results

One hundred and eighty-eight newly arrived laborers participated in this study; out of them 103 (54.8%) were females, 85 (45.2%) were males, and 55.3% were married. The mean age was 26.08 ± 2.67 years. Among those, 86 (45.7%) were illiterate, followed by 60 (31.9%) with elementary education. Most of the participants were housemaids (52.1), drivers (32%), and shepherds (7.4%). The majority of the newly arrived laborers (65%), were from six Asian countries (Philippine, India, Bangladesh, Indonesia, Cambodia, Sri Lanka), while the remaining 35% were from six Africans countries (Sudan, Morocco, Egyptian, Nigeria, Ghana, Somalia) (Table 1).

Table 1 Socio-demographic characteristics of the newly arrived laborers.

This table shows the gender, mean age, education level, occupation, and marital status of all laborers. For details of infections see Tables 2–3.

Variable	No. (%)	
Gender		
Female	103 (54.79%)	
Male	85 (45.21%)	
Mean age (years)	26.08 ± 2.67	
Education level		
Illiterate	86 (45.74%)	
Elementary	60 (31.91%)	
Intermediate	38 (20.21%)	
Secondary	4 (2.13%)	
Occupation		
Housemaid	98 (52.13%)	
Driver	60 (31.91%)	
Shepherd	14 (7.45%)	
General	6 (3.19%)	
Housekeeper	4 (2.13%)	
Babysitter	1 (0.53%)	
Cook	1 (0.53%)	
Cleaner	1 (0.53%)	
Nationality		
Pilipino	56 (29.79%)	
Indian	45 (23.94%)	
Sudanese	28 (14.89%)	
Moroccan	22 (11.70%)	
Bangladeshi	13 (6.91%)	
Egyptian	10 (5.32%)	
Indonesian	7 (3.72%)	
Nigerian	3 (1.60%)	
Cambodian	1 (0.53%)	
Ghanaian	1 (0.53%)	
Somalian	1 (0.53%)	
Sri Lankan	1 (0.53%)	
Marital status		
Married	104 (55.32%)	
Not married	84 (44.68%)	

Out of the 35 infected cases, 26 (74%) revealed single infections, eight (23%) with double infections, and only one (3%) triple infection of different intestinal parasites. Among those infected cases, 19 were infected with STHs (seven from Philippine, five from India, two from Morrocco, one from Indonesia, one from Bangladesh, one from Sudan, one from Egypt, and one from Nigeria) (Table 2, Fig. 1).

There were no significant differences regarding infection with STHs and gender, age, marital status, education level, and residence (p > 0.05). However, higher rates of STHs infection were detected among females, unmarried of less than 26 years old, illiterate and resident in rural areas (Table 3). Similarly, there was no significant variation of infection with STHs and the occupation and the geographic area of the newly arrived laborers (p > 0.05), but higher infection rates were detected among Asians and housemaids (Table 3).

Table 2 The distribution of all detected intestinal parasites including STHs.

This table illustrates the parasitic infection based on the nationality, gender, occupation and education level of each infected laborer.

Type of infection	Detected parasites	No. positive cases	Nationality	Gender	Occupation	Education level	
Single (26)	A. duodenale 1 , 2	1	Pilipino	Female	Housemaid	Intermediate	
A. lumbricoides 1 , 2	4	Indian	Male	Driver	Intermediate	
Nigerian	Male	Housekeeper	Elementary	
Pilipino	Female	Housemaid	Illiterate	
Sudanese	Male	Driver	Elementary	
B. hominis 1	1	Pilipino	Female	Housemaid	Elementary	
E. coli 1	7	Somalian	Female	Housemaid	Elementary	
Sri Lankan	Male	Driver	Illiterate	
Pilipino	Female	Housemaid	Illiterate	
Bangladeshi	Male	Worker	Illiterate	
Indian	Male	Driver	Elementary	
Pilipino	Female	Housemaid	Elementary	
Moroccan	Female	Housemaid	Illiterate	
E. histolytica 1	1	Indian	Male	Driver	Intermediate	
E. nana 1	1	Indian	Male	Driver	Illiterate	
G. lamblia 1	2	Pilipino	Female	Housemaid	Illiterate	
Pilipino	Female	Housemaid	Intermediate	
H. heterophyes 1	1	Indian	Female	Housemaid	Illiterate	
H. nana 1	1	Sudanese	Female	Babysitter	Secondary	
N. americanus 1 , 2	2	Indonesian	Male	Driver	Illiterate	
Bangladeshi	Female	Housemaid	Illiterate	
S. stercoralis 1 , 2	2	Egyptian	Male	Driver	Intermediate	
Pilipino	Female	Housemaid	Illiterate	
T. trichiura 1 , 2	3	Pilipino	Female	Housemaid	Illiterate	
Moroccan	Female	Housemaid	Illiterate	
Indian	Male	Driver	Illiterate	
Double (8)	N. americanus1,2 +B. hominis1	1	Indian	Male	Driver	Illiterate	
N. americanus1,2 + E. histolytica1	1	Pilipino	Female	Housemaid	Illiterate	
E. coli1 + E. histolytica1	1	Pilipino	Female	Housemaid	Illiterate	
E. histolytica1+ B. hominis1	1	Moroccan	Female	Housemaid	Elementary	
S. stercoralis1,2 + A. duodenale1,2	1	Moroccan	Female	Housemaid	Illiterate	
T. trichiura1,2 +N. americanus1,2	2	Indian	Female	Housemaid	Illiterate	
Pilipino	Female	Housemaid	Intermediate	
T. trichiura1,2 + S. stercoralis 2	1	Pilipino	Female	Housemaid	Intermediate	
Triple (1)	T. trichiura1,2 + E. coli1 + I. bütschlii1	1	Indian	Male	Driver	Illiterate	
Notes.

1 Using light microscopy.

2 Using real-time PCR.

As shown in Table 3, none of the investigated newly arrived laborers is aware about the mode of infection of STHs, and there was a significant correlation between the infection among newly arrived laborers that used to walk on soil barefoot in their home countries before arriving to Jeddah.

Discussion

As we mentioned in our previous work, to the best of our knowledge, this study is the first in Jeddah and Saudi Arabia to estimate STHs prevalence among newly arrived expatriate laborers using macroscopic and several microscopic examinations, in addition to real-time PCR for detection and speciation of STHs (Al-Refai & Wakid, 2024). Quantitative PCR is highly sensitive and has rapid utility in detecting STHs (Azzopardi et al., 2021). In our study, the prevalence of STH was 10.11% among 188 newly arrived laborers from different Asian and African countries. Previous studies conducted in Saudi Arabia reported various prevalence values of STHs in Baharah, Makkah, Al-Madina, Al-Baha and Al-Khobar (Wakid, 2020b; Ahmed et al., 2015; Mohammad & Koshak, 2011; Abahussain, 2005). All these previous studies agree with our findings that the participants’ age range between 20–35 years, most of them were females from Asian countries. The predominant female participants were expected because of the large numbers working as housemaids.

Figure 1 Infection with STHs detected among participants belonging to different nationalities.

The total number and the infected cases with STHs among each nationality of the newly arrived expatriate laborers.

Among the detected STHs, 15 cases were infected with one STHs including A. lumbricoides, N. americanus, T. trichiura, four cases each, two cases S. stercoralis and one case A. duodenale. On the other hand, four cases were infected with two STHs (2 cases with T. trichiura and N. americanus; 1 case with S. stercoralis and A. duodenale; 1 case with T. trichiura and S. stercoralis). In the current study, there was no significant variation in the rate of infection with the STHs and sample consistency (p > 0.05), however higher rates of STHs infection were detected in soft/loose samples. Although there was no significant association between STHs infection and education level (p > 0.05), higher infection rates were detected among illiterates (14%) compared to other educational levels. We believe illiterate communities lack of adequate knowledge of proper hygiene and sanitation practices (Sah et al., 2015; Rayamajhi et al., 2014).

Table 3 Infection with STHs and different variables among the newly arrived laborers.

This table illustrates RR and P value of infected laborers with STHs in relation to different variables.

Variables	Categories (N)	Positive STHs n (%)	Negative STHs n (%)	RR (95% CI)	P-value	
Gender	Male (85)	8 (9.41)	77 (90.59)	1.135 (0.478–2.693)	0.777	
Female (103)	11 (10.68)	92 (89.32)	Ref		
Marital status	Married (104)	7 (6.73)	97 (93.27)	0.471 (0.194–1.144)	0.096	
Not married (84)	12 (14.29)	72 (85.71)	Ref		
Geographic region	Asian (123)	14 (11.38)	109 (88.62)	0.676 (0.256–1.793)	0.431	
African (65)	5 (7.69)	60 (92.31)	Ref		
Age group(years)	20–25 (82)	12 (14.63)	70 (85.37)	Ref		
26–35 (106)	7 (6.60)	99 (93.40)	2.216 (0.913–5.377)	0.078	
Education level	Illiterate (86)	12 (13.95)	74 (86.05)	0.853 (0.322–2.264)	0.749	
Elementary (60)	2 (3.33)	58 (96.67)	3.751 (0.727–17.544)	0.117	
Intermediate/Secondary (42)	5 (11.9)	37 (88.1)	Ref		
Awareness about STHs	Aware (0)	0 (0)	0 (0)	-	-	
Not aware (188)	19 (10.11)	169 (89.89)	
Walk on soil barefoot	Yes (69)	12 (17.39)	57 (82.61)	0.338 (0.140–0.819)	0.016	
No (119)	7 (5.88)	112 (94.12)	Ref		
Wash vegetables/ fruits before eating	Yes (66)	6 (9.09)	60 (90.91)	Ref		
No (122)	13 (10.66)	109 (89.34)	0.853 (0.340–2.140)	0.729	
Residence	Rural (128)	11 (8.59)	117 (91.41)	1.552 (0.658–3.658)	0.315	
Urban (60)	8 (13.33)	52 (86.67)	Ref		
Occupation	Housemaid (98)	11 (11.22)	87 (88.78)	Ref		
Driver (60)	7 (11.67)	53 (88.33)	0.962 (0.395–2.346)	0.932	
Other duties (30)	1 (3.33)	29 (96.67)	3.367 (0.453–25.028)	0.235	
Stool consistency	Soft/Loose (115)	10 (8.70)	105 (91.30)	1.418 (0.605–3.322)	0.421	
Formed (73)	9 (12.33)	64 (87.67)	Ref		

This is comparable to findings from another study by Younes et al. (2021), which showed that (7.60%) of illiterate participants represent the highest prevalence of STHs infection compared to other educational levels and was linked to improper hygiene practices among participants families in Qatar. Furthermore, a study done by Abbaszadeh Afshar et al. (2020), in Iran reported no significant relationship between education level and parasitic intestinal infections. The present study revealed no significant difference in positive cases of STHs between Asian and African newly arrived laborers (p > 0.05). Out of the total infected cases, (14/123; 11.38%) were Asians, while (5/65; 7.70%) were Africans. The higher infection rate among Asians agrees with Ahmed et al. (2015) study in Makkah, which showed that 34.50% were Asians, followed by Africans (26.80%). This finding can be explained by the high prevalence of STHs infection in Asian regions compared to African (Werkman et al., 2020). The burden of STHs infection in 118 countries was 70% from the Asian continent, and 26.40% of the samples had at least one helminth species (Pullan et al., 2014). The high burden in Asia is the moist and tropical climate, favorable for STHs survival and transmission. In addition, poor sanitation and hygiene and lack of clean drinking water are essential factors (Utzinger et al., 2010).

In this study regarding the prevalence of STHs among the nationality of the infected participants, the Philippines had the highest number of positive cases (12.50%), followed by Indians (11.10%) and Moroccans (9.10%). This finding is in agreement with a study conducted in Qatar that showed highest STHs prevalence among participants from the Philippines, followed by India and the lowest from Africa (Abu-Madi, Behnke & Ismail, 2008). The proportion of foreign laborers in Saudi Arabia also helps to explain the findings pointing to the relationship between nationality and the prevalence of helminthic parasites. Statistics from the Saudi Arabia migration profiles show that Indians and the Philippines make up the top five migrant communities.

As of 2023, Indians were the second-highest expatriate laborers in Saudi Arabia with a total number of 2.4 million, while the number of the Philippines was 0.73 million (GlobalMedia Insight, 2024). The high numbers of the low-skilled laborer force also make them more susceptible to infection and transmitting STHs. Most of the infected samples came from housemaids and drivers, showing that the natures of their workplace them at a greater risk of transmitting the parasites. These findings were aligned with those reported by Haouas et al. (2021), where the prevalence of parasitic intestinal infections among foreign housemaids in northwestern Saudi Arabia was significantly high. Lack of proper hygiene practices and close contact with maids and drivers put children and the elderly at greater risk of direct transmission of intestinal parasites (Haouas et al., 2021).

Conclusion

Infection with STHs is a significant public health issue to healthcare systems worldwide. Communities must be educated on hygiene practices, and the severity of such parasites to human health as drivers and housemaids come to close contact with families, including children and elderlies. This may put family members at risk, especially as most newly arrived laborers were illiterate, lacking basic hygiene knowledge and practices. We recommend the official authority revise the standard screening tests for newly arrived laborers to minimize the chances of the spread of STHs and other parasites from those laborers to the public.

Supplemental Information

Supplemental Information 1 Raw data related to the workers and the performed tests

Supplemental Information 2 STROBE checklist

Thanks are extended to all the associated personnel in any reference that contributed in/for the purpose of this research.

Additional Information and Declarations

Competing Interests

Author Contributions

Human Ethics

Data Availability

The authors declare there are no competing interests.

Majed H. Wakid conceived and designed the experiments, performed the experiments, analyzed the data, prepared figures and/or tables, authored or reviewed drafts of the article, and approved the final draft.

Mohammad F. Al-Refai conceived and designed the experiments, performed the experiments, analyzed the data, prepared figures and/or tables, and approved the final draft.

The following information was supplied relating to ethical approvals (i.e., approving body and any reference numbers):

Ethics and research committee of Faculty of Applied Medical Sciences, King Abdulaziz University, Jeddah, Saudi Arabia (FAMS-EC2018-012).

The following information was supplied regarding data availability:

The raw data are available in the Supplementary File.

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
