# Peer review of "Contribution of socio-demographic factors in prevalence of soil-transmitted helminth infections among newly arrived laborers in Jeddah, Saudi Arabia"

_PeerJ, doi:10.7717/peerj.18216_

## Round 0.1 · original submission · Major Revisions

I have completed my evaluation of your manuscript. The reviewers recommend reconsideration of your manuscript following major revision. I invite you to resubmit your manuscript after addressing the comments below. When revising your manuscript, please consider all issues mentioned in the reviewers' comments carefully: please outline every change made in response to their comments and provide suitable rebuttals for any comments not addressed. Please note that your revised submission may need to be re-reviewed.

Reviewer 1 ·

Basic reporting

In this manuscript, the authors examine the relationship between socio-economic conditions and the prevalence of helminth infections (STHs) among newy-arrived laborers in Jeddah, Saudi Arabia. The writing is clear, the data are well-organized, and the topic is significant for public health

Experimental design

Although the authors referenced their previous study (Al-Refai & Wakid, 2024) in the methods section, it is important to provide more details in this manuscript. For instance, how were the participants selected? Were there any participants excluded? How were the procedures such as macroscopic examination, direct wet smears, sedimentation formal-ether technique, permanent staining with trichrome and modified Kinyoun's, rapid immunochromatographic diagnostic tests, and real-time PCR for STHs performed? What were the primer sequences used, and so on?

Validity of the findings

1. I suggest the authors also provide information about the length of time the participants have stayed in Jeddah since their first arrival and explore whether there is a relationship between this factor and the incidence of STH cases

2. One of the factors investigated in this study is the education level. However, the authors should better define what is meant by ‘elementary’, ‘intermediate’, and ‘secondary.' Since the journal's audience is international and different countries have varying education systems, it is important to provide clear and detailed explanations.

3. Lines 119 to 122 contain a confusing sentence. I suggest the authors revise this sentence for clarity.

Reviewer 2 ·

Basic reporting

no comment

Experimental design

no comment

Validity of the findings

no comment

Additional comments

Dear authors,
The manuscript is easy to read and understand.
It is clear, that the methodology fulfills the aims, the results are well described and the discussion is concise.
Overall, the manuscript is good and helps us understand a little better the role of migration in the dispersal of STH.
I recommend adjusting the title because I could not find data to sustain the socio-economic evaluation just socio-demographic.

Reviewer 3 ·

Basic reporting

1. This study reported about the prevalence of STH and factors that might be associated with the infection among migrant workers. The title of this manuscript emphasizes STH infection, but the first narrative is on overall intestinal infections, including protozoa and cestodes? The terminology used to describe the studied population should be consistent throughout the manuscript. The author uses the term “expatriate labor’’ in the title but there is inconsistency within the text, where the terms “worker”, “expatriate”, “foreign worker” are used interchangeably. It’s important to note that "expatriate" typically refers to skilled professionals working abroad. Please consider whether "newly arrived migrant workers" might be a more appropriate term for this study. Once the appropriate term is selected, please also ensure it is used consistently throughout the manuscript.
2. Abstract- Nineteen cases of STH. Percentage? Revise the result section in the abstract. Your title emphasizes STH infection, but your focus is on overall intestinal infections, including protozoa and cestodes?
3. Introduction- Please check the flow of the introduction. For example, line 70-72 should be combined in paragraph 2. In the conclusion- the author recommended about “the official authority revise the standard screening tests for newly arrived workers to minimize the chances of the spread of STHs and other parasites. Suggest to include in the introduction- current screening tests for newly arrived workers? What screening included in current procedure?

Experimental design

4. Materials and method- My major concern in this section is the limited description of how the study participants were recruited? The author only mentions at line 100, “This cross-sectional study included newly arrived expatriate laborers to Jeddah”. It would be beneficial for the author to provide more detail on the recruitment process. Specifically, what criteria were used to define "newly arrived"? For example, does this refer to individuals who arrived within the past month?or past few days? Additionally, how long was the period for collecting stool specimens in this study? Suggest to indicate the time frame of sample collection. Where were the participants recruited? Any inclusion and exclusion criteria?
5. Materials and method- The author also mentions “a verbal interview was conducted for each participant”. Considering the participants are from many countries, what language was used during the interview?
6. Line 108- Please check your sentence.
7. For data analysis, why relative risk (RR) calculation was chosen instead of logistic regression analysis? The reason why I ask this information since some of the variables in Table 3 have more than 2 categories (education level and occupation). Relative Risk is most commonly used when you have a binary outcome (e.g., disease/no disease). If your outcome variable has more than two categories, RR is not typically used because it is designed for binary outcomes. I suggest you to collapse or recode the outcome into a binary format (e.g., grouping several categories together) if you still want to use RR. All these information is crucial for understanding the context of the findings and ensuring the study's methodology is clear.

Validity of the findings

8. Results- The results section needs revision. Table 2 is too descriptive, making it unclear what key information the author is trying to convey? I suggest moving this table to the supplementary materials to streamline the main text and focus on the most relevant findings. Perhaps the author can start with overall prevalence, followed by percentage according to the type of infection (the main causative agents to the least causative species) and multiparasitism status. What is the severity of the STH infection? Just wondering, since most of the infected participants working as housemaid and driver, did the author provide medication to the infected participants after the study? Is there any data on monthly income status? Is there any relationship with this variable with the infection?
9. Higher rates of STHs infection were detected among workers used to walking on soil barefoot. Does this refer to workers who walked barefoot in their home countries before arriving? Are they still walking barefoot in their current environment in Saudi? Clarifying this distinction would strengthen the interpretation of the findings.

Additional comments

10. Standardize the number of decimal points throughout the manuscript.

---

## Round 0.2 · Minor Revisions

Dear Author,
Kindly fulfill reviewers' queries properly and submit the revised manuscript with track changes or highlights.

Reviewer 2 ·

Basic reporting

no comment

Experimental design

no comment

Validity of the findings

no comment

Additional comments

no comment

Reviewer 3 ·

Basic reporting

There have been significant improvements in the manuscript. However, there are still a few questions that have not been addressed by the authors.
1) Please change the terminology from "labors" to "laborers" (--newly arrived expatriate laborers). Laborer- a person who does the labor work.
2) I suggest including the following in the introduction: What are the current standard screening tests for newly arrived workers? What screening tests are included in the current government procedures? Perhaps, based on this study, certain policies can be strengthened, particularly in testing for intestinal parasites, many of which are food-borne.
3) It would be helpful to clarify the severity of the STH infection among the laborers.
4) Since the author mentioned that, stool samples were checked for consistency. How many of them had loose stool/diarrhea during the sample collection?

Experimental design

As above.

Validity of the findings

As above.

Additional comments

Other comments can be found in the track changes manuscript.

Annotated reviews are not available for download in order to protect the identity of reviewers who chose to remain anonymous.

---

## Round 0.3 · accepted · Accept

It is a pleasure to accept your manuscript entitled "Contribution of socio-demographic factors in prevalence of soil-transmitted helminth infections among newly arrived laborers in Jeddah, Saudi Arabia" in its current form for publication in PeerJ.